# CSD-Grown Y_1−*x*_Gd*_x_*Ba_2_Cu_3_O_7−*δ*_-BaHfO_3_ Nanocomposite Films on Ni5W and IBAD Technical Substrates

**DOI:** 10.3390/nano10010021

**Published:** 2019-12-19

**Authors:** Pablo Cayado, Hannes Rijckaert, Manuela Erbe, Marco Langer, Alexandra Jung, Jens Hänisch, Bernhard Holzapfel

**Affiliations:** 1Karlsruhe Institute of Technology (KIT), Institute for Technical Physics (ITEP), Hermann-von-Helmholtz-Platz 1, 76344 Eggenstein-Leopoldshafen, Germany; manuela.erbe@kit.edu (M.E.); marco.langer@kit.edu (M.L.); alexandra.jung@kit.edu (A.J.); jens.haenisch@kit.edu (J.H.); bernhard.holzapfel@kit.edu (B.H.); 2Department of Chemistry, Sol-gel Centre for Research on Inorganic Powders and Thin Films Synthesis (SCRiPTS), Ghent University, Krijgslaan 281-S3, 9000 Ghent, Belgium; Hannes.Rijckaert@UGent.be

**Keywords:** CSD, Ni5W, IBAD, YGBCO, BHO, nanocomposites

## Abstract

Chemical solution deposition (CSD) was used to grow Y_1−*x*_Gd*_x_*Ba_2_Cu_3_O_7−δ_-BaHfO_3_ (YGBCO-BHO) nanocomposite films containing 12 mol% BHO nanoparticles and various amounts of Gd, *x*, on two kinds of buffered metallic tapes: Ni5W and IBAD. The influence of the rare-earth stoichiometry on structure, morphology and superconducting properties of these films was studied. The growth process was carefully studied in order to find the most appropriate growth conditions for each composition and substrate. This led to a clear improvement in film quality, probably due to the reduction of BaCeO_3_ formation. In general, the superconducting properties of the films on Ni5W are significantly better. For *x* > 0.5, epitaxial ~270 nm thick YGBCO-BHO films with *T*_c_ > 93 K and self-field *J*_c_ at 77 K ~2 MA/cm² were obtained on Ni5W. These results highlight the potential of this approach for the fabrication of high-quality coated conductors.

## 1. Introduction

Most of the present-day research in the field of applied superconductivity is devoted to the improvement of second-generation superconducting tapes, so-called coated conductors (CCs) [1,2,3,4], whose base materials are the *RE*Ba_2_Cu_3_O_7−δ_ (*RE*BCO, *RE:* rare earth) compounds. These materials present outstanding properties, such as their high critical temperature (*T*_c_) and high current carrying capability at high magnetic fields that make them suitable for several power and high-field applications.

Among the *RE*BCO group, YBa_2_Cu_3_O_7−δ_ (YBCO) is certainly the best-known compound. However, multiple publications suggest alternative *RE*BCO compounds to outperform YBCO in several ways [5,6,7,8,9,10]. Differences in the electronic structures, valence states and ionic radii of the *RE* atoms are responsible for these improvements [11,12,13,14]. The synthesis of some of these compounds is much more difficult than the YBCO though because of the different *RE*^3+^ ion size. On the one hand, larger *RE*^3+^ ions, like Nd^3+^ or Eu^3+^, have certain tendency to partially substitute Ba^2+^ ions. On the other hand, the smaller ions, such as Yb^3+^ or Er^3+^, do not fit appropriately into their corresponding lattice positions creating vacancies. Both cases lead to a drastic decrease of the stability of the *RE*BCO phase [5,11,15,16].

A simple possibility to overcome some of the complications in the synthesis of alternative single-*RE-*BCO compounds but yet to benefit from the improvement of the superconducting properties is to mix different *RE*^3+^ ions. In this regard, we recently reported on the possibility to enlarge the processing window for epitaxial high-quality films of Y_1−*x*_Gd*_x_*Ba_2_Cu_3_O_7−δ_-BaHfO_3_ (YGBCO-BHO) on single crystal substrates with respect to the corresponding single-*RE* films [17]. Various combinations of *RE*^3+^ ions, including the one used in this work, such as Y/Sm [14,18,19], Y/Gd [19,20], Y/Pr [21,22,23], Y/Eu/Gd [24], and Nd/Eu/Gd [25,26] were also investigated by several groups and revealed better superconducting transport properties due to an enhancement of the flux pinning by the *RE* mixing itself. It was proposed that the improvement of the superconducting properties was due to the alteration of the properties in very confined areas in the surroundings of the dopants which increases the pinning force densities of the system.

In general, substitution on the *RE* sites of the crystal structure could produce nanometric or more macroscopic structures that could be partially or totally non-superconducting, depending on several variables such as the substituent, its molar fraction, and others [27,28]. Therefore, the improvement of the superconducting properties is related to the nature of the *RE* dispersion. If the *RE* elements are homogeneously dispersed in the structure, *T*_c_ will be influenced in very localized areas in the vicinity of the doped areas. Such fluctuations of *T*_c_ are expected to improve the pinning properties. If clusters with accumulations of one sort of *RE* are formed, microscopic strain will be generated in their surroundings at the same time, which would also contribute to flux pinning in the films [27].

Apart from the benefits of *RE* mixing, and in order to improve the pinning properties in such films further, secondary phases can be included in the matrix to produce *RE*BCO nanocomposites. This topic has been extensively studied by many groups evidencing that the in-field transport properties of *RE*BCO films can be extensively enlarged by using both the “in situ” and the “ex situ” approaches [29,30,31,32,33,34,35,36].

In our previous study, the interplay between and combined effects of *RE* mixing and nanoparticle inclusion on YGBCO-BHO films deposited on single crystals were investigated [17]. In order to transfer these results to long-length production, buffered metallic tapes were used as substrates for stationary deposition as an intermediate step. Here, chemical solution deposition (CSD), a scalable and low-cost method for the fabrication of CCs [37,38,39,40], and, specifically, the TFA-MOD route [41], has been used to grow YGBCO-BHO nanocomposite films by the “in situ” approach on two different technical substrates: Ni5W and IBAD. Y^3+^ and Gd^3+^ ions were chosen for the mixed phase because this combination has already been prepared in the past by pulsed laser deposition (PLD) or metal-organic chemical vapor deposition (MOCVD) with excellent properties both on single crystals [24] and buffered metallic tapes [42,43,44,45,46]. CSD has already shown promising properties for YGBCO films [19,20,21,22,23,24], but still the effort is not comparable to YBCO or GdBCO. Moreover, although the Ni5W and IBAD substrates have been widely used for preparing YBCO film by different techniques [47,48,49,50,51,52,53], the growth of GdBCO or YGBCO films on such templates has not been widely investigated. Here, we investigate the influence of morphology, texture quality, and CeO_2_ buffer layer thickness of these templates on the microstructure and superconducting properties of Y_1−*x*_Gd*_x_*Ba_2_Cu_3_O_7−δ_-BaHfO_3_ nanocomposite films with different Gd content *x*.

## 2. Sample Preparation and Characterization Techniques

### 2.1. Sample Preparation

The TFA solutions used in this work were prepared following the protocol reported in Ref. [9]. Two batch solutions of YBCO with 12 mol% BHO and GdBCO with 12 mol% BHO (12% BHO from now on) were prepared by weighing and mixing the acetates of Y or Gd, Ba and Cu (purity > 99.99%, Alfa Aesar, Kandel, Germany) in a ratio of 1:2:3 together with hafnium(IV) 2,4-pentanedionate (Hf(acac)_4_, 97+%, Alfa Aesar) in deionized water. After that, trifluoroacetic acid (TFAH, 99.5+%, Alfa Aesar, Kandel, Germany) was added to transform the acetates into trifluoroacetates. The obtained solutions were then purified from the initial solvent water and other impurities using a rotary evaporator in vacuum obtaining a highly viscous residue that was re-dissolved in absolute methanol (99.9%). In order to reduce undesired remains of water further, this process was repeated several times. The desired concentration of 0.25 mol L^−1^ in Y or Gd was adjusted by adding anhydrous methanol, obtaining a dark blue solution. Finally, the YGBCO + 12%BHO solutions were prepared by mixing the batch solutions in different ratios to obtained the *RE* stoichiometries aimed for.

The YGBCO + 12%BHO films with an approximate thickness of 270 nm were produced, at first, by depositing the precursor solutions on 10 × 10 mm² Ni5W and IBAD substrates by spin coating (6000 rpm for 30 s). The fully CSD-buffered Ni5W substrates were supplied by Deutsche Nanoschicht GmbH and consisted of a thin (10–12 nm) top layer of CeO_2_ on three layers of 100 nm each one of La_2_Zr_2_O_7_ (LZO) that were, in turn, deposited on top of a biaxially textured Ni-W alloy substrate. The IBAD substrates, from SuperOx, had the following stacking sequence: PLD-grown 100–200 nm thick CeO_2_ on magnetron-sputtered 30–50 nm LaMnO_3_ on 5–7 nm IBAD-MgO on 30–50 nm Y_2_O_3_ or LaMnO_3_ and 50 nm Al_2_O_3_ deposited on a ~60 µm thick non-magnetic Hastelloy C276 substrate.

The later “standard” pyrolysis and growth processes were detailed in Ref. [10]. The changes made in the “standard” growth process that were employed in this work will be explained in detail later.

### 2.2. Thin-Film Characterization

The optical images of the pyrolyzed films were taken by a Keyence VHX-1000 digital microscope with motorized *z*-axis. The microstructure and phase purity of the films were analized by X-ray diffraction (XRD) using a Bruker D8 diffractometer with CuK_α_ radiation in Bragg-Brentano geometry and on a Rigaku SmartLab 3kW (40 kV, 30 mA) *5*-axis diffractometer with a goniometer radius of 300 mm, parallelized Cu-K_α_ radiation and a HyPix-3000 2D detector. The surface morphology was studied by a LEO 1530 scanning electron microscope (SEM) with field-emission gun (0.1 kV and 30 kV) by Zeiss. The self-field critical current density, $J_{c}^{\mathrm{sf}}$, at 77 K was measured inductively with a Cryoscan (Theva, 50 µV criterion). The critical temperature *T*_c_ was evaluated by transport measurements in 4-point geometry at a 14-T Quantum Design Physical Property Measurement System (PPMS). *T*_c_ is defined as *T*_c,90_, i.e., the temperature at which the resistance is 90% of the value above the transition.

## 3. Results and Discussion

The first requirement for films with good superconducting properties is a homogeneous and defect-free layer after the pyrolysis process, which was achieved for both substrates (black spots in the IBAD image come from some dirty in the lens of the microscope), as illustrated in Figure 1.

On the other hand, the optimal growth conditions for the different YGBCO-BHO films depend on the amount of Gd, *x* [17]. When using SrTiO_3_ as a substrate, the optimal crystallization temperature (*T*_crys_) increases from 780 °C for *x* = 0 to 810 °C for *x* = 1 and the optimal oxygen partial pressure (*p*O_2_) reduces from 200 ppm for *x* = 0 to 50 ppm for *x* = 1. Here, those optimized *p*O_2_ values have been used with regard to *x* but, as a consequence of progressing substrate deterioration at high temperatures, the optimal temperatures on metallic tapes had to be adapted, e.g., they decreased to 770 °C for *x* = 0 and 790 °C for *x* = 1 (Table 1).

The XRD patterns in Figure 2a,b show intense (00*l*)YGBCO reflections in all the films grown on both types of substrates (except for *x* = 0 on IBAD), which indicates a strong preference for the *c*-axis orientation. The areal intensity of these peaks, and especially of (005) which has roughly the same theoretical relative intensity for YBCO and GdBCO, increases with Gd content *x*. Since the film thickness (~270 nm) does not show a clear trend and the *c*-axis texture quality is decreasing with *x* (from 2D XRD measurements, not shown here for all *x* values), an increasing *RE*BCO phase formation with increasing Gd content *x* can be concluded. The Full width at half maximum (FWHM) of these peaks is slightly decreasing with *x*, indicating qualitatively decreasing strain and/or increasing coherently scattering volume with *x*. Furthermore, nearly no (103)YGBCO or (*h*00)YGBCO reflections associated to randomly or *a-b*-oriented grains, respectively, are detected apart from a minor (200)YBCO peak in the YBCO film on Ni5W.

These structural characteristics seem to be linked to the amount of BaCeO_3_ (BCO) formed in the films or more precisely at the substrate interface. BCO, a product of the reaction of the CeO_2_ top buffer layer with the Ba from the *RE*BCO matrix, occurs in all the films, but seems to decrease with the Gd content *x* considering especially the (002) reflection in the *θ*-2*θ* scans (Figure 2). This BCO-*x*-dependence appears not as strong in the IBAD samples due to a more polycrystalline growth of BCO as revealed by 2D XRD images (Figure 3). It seems very likely that BCO forms, in the presence of high HF concentrations, through a reaction of intermediary BaF_2_ with CeO_2_ and, therefore, prior or parallel to the formation of *RE*BCO. For Y, the two reaction paths towards YBCO and BCO are thermodynamically equally likely and, therefore, the reaction rate towards YBCO has to be enhanced kinetically by setting favourable conditions since BCO forming after the YBCO formation is not harmful [54]. If Y is exchanged with more and more Gd, the reaction towards *RE*BCO seems to be enhanced over the BCO formation.

Two effects may simultaneously lead to this tendency: first, the samples with different Gd content *x* are grown at different deposition conditions, Table 1. Whereas the temperature differences of 10–20 °C are negligible, the differences in optimal oxygen partial pressure are considerable. In fact, the areal intensity of (002)BCO on Ni5W increases nearly exponentially with *p*O_2_ (except for *x* = 1 which is most likely due to the suboptimum growth temperature). Very likely and plausibly, the formation rate of BCO from CeO_2_ and Ba species depends strongly on *p*O_2_. The second possible reason for this reduced amount of BCO is a different reaction path towards the *RE*BCO phase. While in the case of YBCO, the formation of the intermediary or competing Ba-free phase Y_2_Cu_2_O_5_ is well-known (some traces of Y_2_Cu_2_O_5_ are often found in the final YBCO films, e.g., on IBAD marked with *) [55], the reaction chain towards GdBCO via TFA-MOD might circumvent the formation of a similar phase Gd_2_Cu_2_O_5_. Just as well, other intermediary phases might be promoted, although no traces have been found in the final films with higher amounts of Gd *x*. Lee et al. for example mention a Ba-rich phase GdBa_6_Cu_3_O*_x_* at low oxygen partial pressures (<20 ppm) [56]. Since the melting point of the corresponding YBa_6_Cu_3_O*_x_* phase is 25 °C higher than for GdBa_6_Cu_3_O*_x_* [57], and YBCO is grown at lower *T* and even higher *p*O_2_, the formation of such intermediary Ba-rich phases is more likely for higher Gd content. In combination, since less Ba is bound in intermediary phases for lower *x*, the transformation of CeO_2_ to BCO is more likely. The exact nature of the reaction path towards TFA-MOD-GdBCO is unclear at present and requires a more profound investigation. What can be concluded, nevertheless, is that the driving force for the formation of BCO seems significantly reduced for larger *x*, either by the associated reduction of necessary oxygen partial pressure or the different reaction path [58].

In any case and especially for low *x*, the formation of BCO needs to be contained or at best entirely avoided. An optimization of the growth parameters would be required in this case. Also, the results of some preliminary experiments carried out in our group suggest that the use of a solution formulation with a reduced amount of fluorine helps to reduce the formation of BCO.

Since linear XRD-*θ*-2*θ* -scans could only clarify the *x*-dependence of the growth behaviour within a substrate series to some extent, but did not give sufficient information about the distinctions between the different substrates, more detailed information about the texture of the films were obtained by two-dimensional XRD *θ*-2*θ* frames (Figure 3). The preferential orientation of the BCO in YBCO + 12%BHO films is clearly different on both substrates. While on Ni5W only the (200) orientation (pseudo-cubical cell, ICSD-No. 29109) can be identified without any doubt and with the substrate-specific texture quality, the same reflection has a ring form on IBAD and co-exists with the even more prominent (110) orientation. The latter is the main reflection of BCO, but seems to have a distinct, but rather poor texture (a ring is suggested, but incomplete). This means that BCO tends to grow epitaxially on the roughly 10 nm-thin CeO_2_ layer of the Ni5W substrate. On IBAD, on the other hand, considerably more BCO is formed, which is certainly due to the thicker CeO_2_ layer, and it grows almost entirely randomly oriented. These differences in orientation, which are probably caused by differences in the reactivity (higher in IBAD than in Ni5W) of both CeO_2_ buffer layers with the *RE*BCO precursors, and the amount of BCO are substantial for the different texture of the YBCO films on both substrates. On Ni5W, the YBCO texture is adapted to the specific texture of the substrate as is the BCO. Some *a-b* grains are present as well. On IBAD, however, YBCO appears completely randomly oriented. Thus, if BCO is formed, *c*-axis nucleation of REBCO is still possible as long as the BCO grows with a (*h*00) texture. Random orientations of BCO disturb the *c*-axis growth of *RE*BCO severely. This tendency is observed likewise for larger values of *x*, although less drastic than in YBCO.

The structural features observed in XRD accord with SEM images of the surface morphology of the films (Figure 4). In general, the films present a considerable degree of porosity on both substrates. Yet, there is a clear trend towards denser films for increasing Gd content. On the other hand, all films except YBCO + 12%BHO are mostly free of visible *a-b* grains (needle-shaped structures forming 90° between each other) or randomly oriented grains (needles oriented in any possible direction) which agrees with the XRD structural data. When BCO is formed, its location at the substrate interface and the large lattice mismatch towards YGBCO leads to the growth of misoriented grains. Therefore, the fact that films with a higher Gd content exhibit less misoriented grains is explained by the lower tendency to form BCO at the interface.

There is a general tendency for both *T*_c_ and inductive $J_{c}^{\mathrm{sf}}$ at 77 K to increase with the Gd content *x*, see Figure 5. Both values are slightly higher on Ni5W than on IBAD at every composition, which is in accordance with the more pronounced formation of randomly oriented BCO on IBAD, as shown in the case of *x* = 0, and the consequent texture deterioration of the *RE*BCO phase. At temperatures close to *T*_c_, *T*_c_ itself has a major influence on *J*_c_ and is, thus, reflected in *J*_c_, unless features in the microstructure predominate. The latter seems to be the case for the GdBCO + 12%BHO film on Ni5W. For this particular composition the overall highest *T*_c_ of ~94.5 K is reached but the maximum $J_{c}^{\mathrm{sf}}$ at 77 K is found at lower *x* = 0.66 with a value of 2 MA/cm². Higher growth temperatures would be required for optimally grown GdBCO + 12%BHO with improved texture, which could not be applied on metallic tapes due to substrate deterioration in the crystallization stage. For YGBCO + 12%BHO films, the processing parameter window is wider and allows the growth of good films even slightly away from the optimum [17]. Furthermore, the improvement in *J*_c_ due to BHO nanoparticles is clearly seen in Figure 5c for the particular case of YBCO films with and without nanoparticles and deposited on Ni5W substrates. Due to the pinning effect of the BHO NPs, *J*_c_ of the nanocomposites is significantly larger than the pristine films.

Although the superconducting properties of the YGBCO + 12%BHO films are promising, the self-field *J*_c_ values are still far below the values achieved on SrTiO_3_ (STO) [17]. Partially, this can be explained by the grain-boundary network in the films on tape [59]. But also, as discussed in detail in refs. [10,11,12,13,14,15,16,17,18,19,20,21,22,23,24,25,26,27,28,29,30,31,32,33,34,35,36,37,38,39,40,41,42,43,44,45,46,47,48,49,50,51,52,53,54,55,56,57,58,59,60], porosity degrades *J*_c_ since pores reduce the cross-section and hinder the current flow. However, the maximum values of *T*_c_ and $J_{c}^{\mathrm{sf}}$ at 77 K in our YGBCO + 12%BHO films are similar to the ones reported previously for YBCO films with similar thickness and deposited by different techniques [50,51,52,53].

## 4. Conclusions

Y_1−*x*_Gd*_x_*Ba_2_Cu_3_O_7−δ_-BaHfO_3_ nanocomposite films were prepared by chemical solution deposition on two technical substrates: Ni5W and IBAD. The influence of the Gd content *x* on the quality of the films on each substrate has been investigated. In general, well-textured films free of misoriented grains have been achieved for *x* > 0.5. These films with higher content of Gd have a lower tendency to form BaCeO_3_, either by the associated reduction of necessary oxygen partial pressure or the different reaction path, which disturbs the desired *c*-axis-oriented growth. They also show higher microstructural density and homogeneity. A remarkable improvement of the superconducting properties of the films with larger *x* has been observed on both types of substrates. However, the *T*_c_ and $J_{c}^{\mathrm{sf}}$ values at 77 K were higher in the films deposited on Ni5W than on IBAD probably due to the larger formation of BaCeO_3_ since the CeO_2_ buffer layer is much thicker for the latter and its formation randomly oriented.

## Figures and Tables

**Figure 1 nanomaterials-10-00021-f001:**
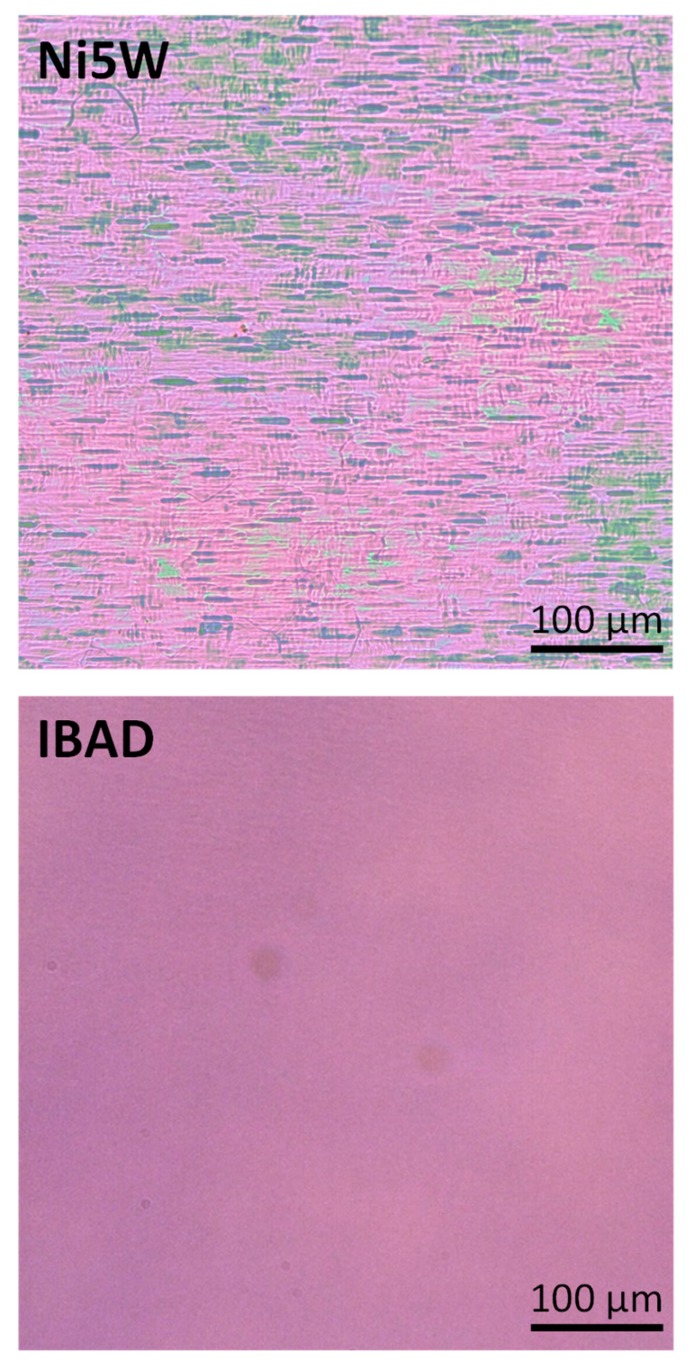
Representative pictures of pyrolyzed films deposited on Ni5W and IBAD substrates. The images were taken by an optical microscope working in reflection mode with white illumination and without polarization filters.

**Figure 2 nanomaterials-10-00021-f002:**
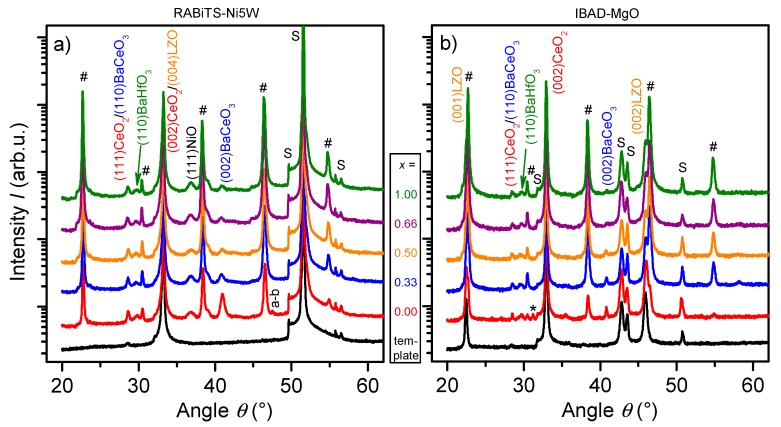
X-ray diffraction (XRD) patterns of Y_1−*x*_Gd*_x_*Ba_2_Cu_3_O_7−δ_+12%BHO films grown on (**a**) Ni5W and (**b**) IBAD substrates. The reflections marked with # come from the YGBCO and the ones marked with S from the substrate and experimental setup, * Y_2_Cu_2_O_5_.

**Figure 3 nanomaterials-10-00021-f003:**
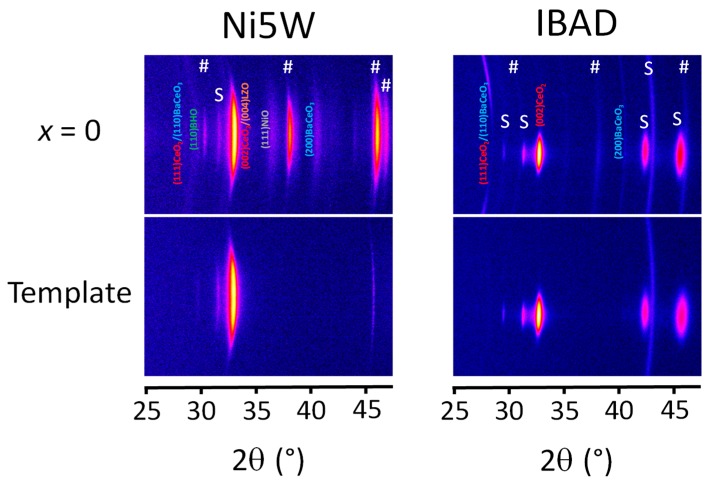
Two-dimensional (2D) XRD *θ*-2*θ* frames of the clean substrates and YBCO + 12%BHO films grown on Ni5W and IBAD substrates. The reflections marked with # come from the YBCO and the ones marked with S from the substrate and experimental setup. The misalignment of the peaks on IBAD samples comes from the inner tilting of the CeO_2_ buffer.

**Figure 4 nanomaterials-10-00021-f004:**
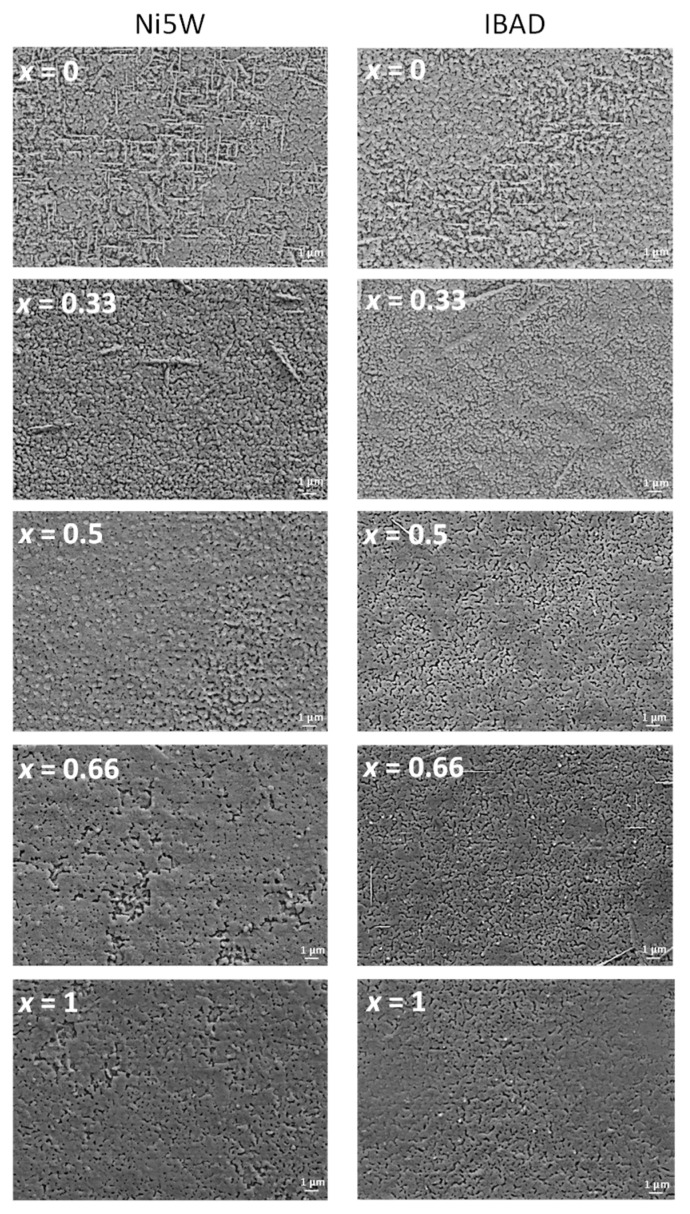
Surface morphology of Y_1−*x*_Gd*_x_*Ba_2_Cu_3_O_7−δ_ + 12%BHO films with different Gd content *x* grown on Ni5W and IBAD substrates and observed via scanning electron microscope (SEM) imaging.

**Figure 5 nanomaterials-10-00021-f005:**
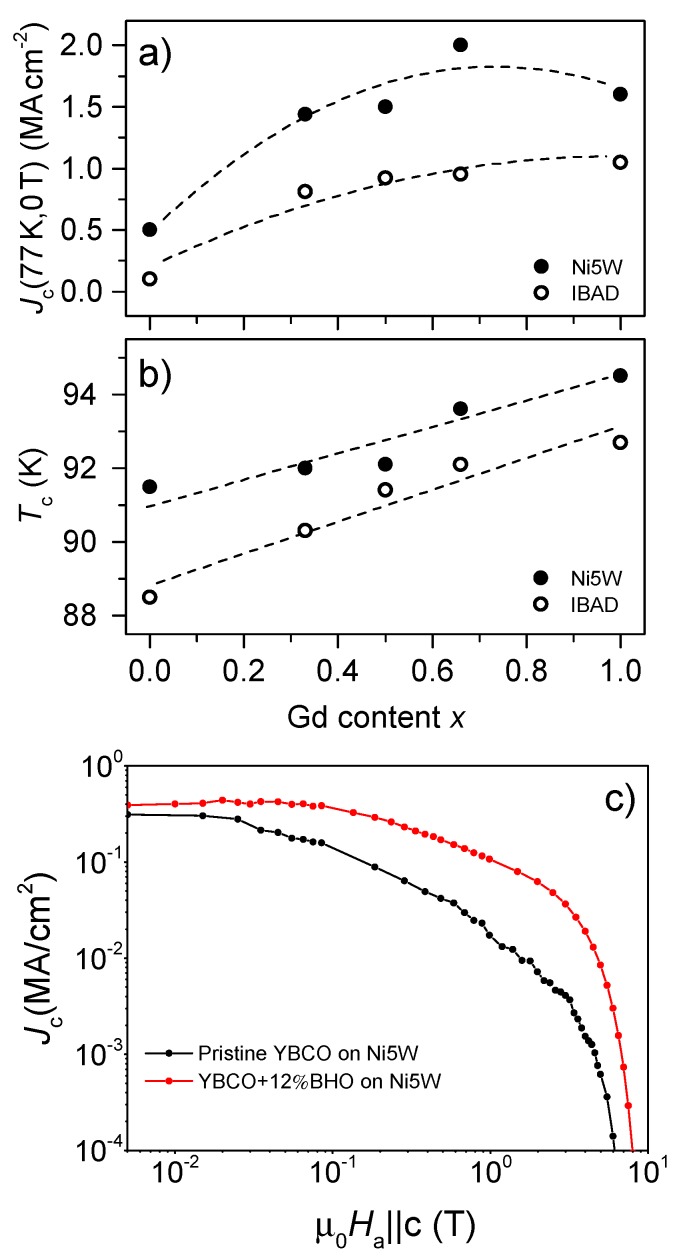
Dependence of the inductive (**a**) $J_{c}^{\mathrm{sf}}$ at 77 K and (**b**) *T*_c_ on the amount of Gd present in the YGBCO + 12%BHO films deposited on Ni5W (close symbols) and IBAD substrates (open symbols). The graph (**c**) shows the magnetic field dependence of *J*_c_ at 77 K for pristine YBCO and YBCO + 12%BHO films deposited on Ni5W.

**Table 1 nanomaterials-10-00021-t001:** Optimized crystallization temperatures (*T*_crys_) and oxygen partial pressures (*p*O_2_) depending on Gd content *x* in Y_1−*x*_Gd*_x_*Ba_2_Cu_3_O_7−δ_ + 12%BHO nanocomposite films on tapes.

*x*	Optimized *T*_crys_ (°C)	Optimized *p*O_2_ (ppm)
0	770	200
0.33	780	150
0.5	780	100
0.66	790	75
1	790	50

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
