# Peer review of "CSD-Grown Y1−xGdxBa2Cu3O7−δ-BaHfO3 Nanocomposite Films on Ni5W and IBAD Technical Substrates"

_nanomaterials, 2019, doi:10.3390/nano10010021_

Round 1

Reviewer 1 Report

This manuscript presents the properties of YGBCO-BHO nanocomposite films on two types of substrates such as Ni5W and IBAD. The developed process for the nanocomposite films has been published in the authors’ previous paper where STO substrates were used (ref 17). This work is focused on the use of two different technical substrates for implementing the developed CSD process of YGBCO-BHO nanocomposite films. The presented results are carefully compared and discussed. It is believed that the presented approach would provide an important option for the industrial production of CCs.

Reviewer 2 Report

The paper investigates the growth of Y1-xGdxBa2Cu3O7-δ-BaHfO3 (YGBCO-BHO) nanocomposite films on two kinds of buffered metallic tapes: Ni5W and IBAD. In particular, the influence of rare-earth stoichiometry on structure, morphology and superconducting film properties has been evaluated.

Overall, papers dealing with this issue are of general interest because a deeper comprehension of the film growth on technical substrates is fundamental to properly optimize the precursor solution composition and process parameters in order to demonstrate the scalability of CSD method. Therefore, the topic is suitable for publication. However some details could be included to improve the quality of the work.

First of all, the authors studied the impact of Y/Gd ratio on film properties, but different growth conditions were adopted for YGBCO-BHO depending on Gd content. In particular, the crystallization temperature and the oxygen partial pressure were chosen on the basis of the optimization study made on single crystal and previously published by the authors (ref. 17). However, these parameters significantly varied depending on x value, but can influence the film properties such as the x value itself. For example, the oxygen partial pressure can be responsible of the different amount of formed BaCeO3 as effectively mentioned by the authors in the Discussion part (lines 166-172). Therefore, I don’t understand why, in other parts of the manuscript, the authors ascribe such difference only to Gd content: lines 185-186 “What can be concluded nevertheless is that the driving force for the formation of BCO seems significantly reduced for larger x”; lines 263-264 “These films with higher content of Gd have a lower tendency to form BaCeO3 which disturbs the desired c-axis-oriented growth”.

Moreover, it is not clear, the reason of different BaCeO3 growth: epitaxial on Ni5W and randomly oriented on IBAD. Did the CeO2 top layers differ only in the thickness? Did the authors perform analysis of in-plane texture of the CeO2 layers?

Regarding the superconducting properties of nanocomposite films, it should be interesting a comparison with other similar studies reported in the literature. In particular, Ni5W and IBAD substrates have been widely used for YBCO films deposited by CSD.

Furthermore, the in field Jc behavior should be shown in order to see if the added nanoparticles are acting as pinning centers.

Other minor points:

Line 14 “A careful optimization of the growth process for each composition and substrate led to a clear improvement in film quality”: the optimization was carried out previously (rif.17). In this work, it seems to be just adapted to the technical substrates. Lines 125-127 “a homogeneous and defect-free layer after the pyrolysis process, which was achieved for both substrates, as illustrated in figure 1”: it is not so evident the defect-free film. Actually, in figure 1, I can see some streaks (Ni5W) and darker circular areas (IBAD). May the authors provide information on the used light (e.g. polarized) and on the stoichiometry, i.e. x value, of the shown films? Line 147: the film thickness value does not correspond to the one reported at line 98. Line 192: is it possible to add in the caption the meaning of * symbol? Figure 3: it should be interesting to add XRD spectra of GdBCO films (x=1). Line 238 “more pronounced formation of BCO on IBAD”: it is unclear if the authors refer to the total amount of formed BCO (both epitaxial and misoriented) or just to the misoriented. In any case, how was quantified such difference? Figure 5: RABITS name used in the legend should be changed into Ni5W in accordance with the other figures. Moreover, any red curves is visible in the figure (as instead mentioned in the caption). Lines 270-273: “However, a reformulation of the employed full-trifluoroacetate solution (all the precursors are trifluoroacetate salts) to other low-fluorine solutions appears to be necessary in order to reduce the overall formation of BaCeO3 and further improve the superconducting properties.” This point should be better explained. How a reduced fluorine content should influence BCO formation?

Round 2

Reviewer 2 Report

I have read the revised version of the article and the authors' answer. Most of the requested details have been supplied. But I still have some concerns.

In the following my doubts are reported:

Regarding the possible effect of other parameters, such as the oxygen partial pressure, on the observed results, the authors clarified this point, as requested, in the discussion part. However, in the conclusion part no text was changed and the observed improvements of film properties still seem to depend only to Gd content, so I think it should be better to add, also in the conclusions, a sentence like the ones introduced at lines 184-189. Moreover, I think it should be recommended to add some references to the sentences used at lines 184-189. For example, the article just published by Chu et al on Crystal Growth and Design Journal (DOI: 10.1021/acs.cgd.9b01120) would be appropriate because is in accordance with the results proposed in the manuscript. The second answer supplied by the authors (regarding BCO orientation) is not supported by data or references. Moreover, no comment was added in the text, neither the possibility that the authors have no explanation about this topic. I asked to add in Figure 3 the 2D-XRD spectra of GdBCO films (x=1), the authors replied me that they performed 2D-XRD analyses only for YBCO (x=0). Therefore, I don’t understand how can they state, at lines 234-235 “Both values are slightly higher on Ni5W than on IBAD at every composition, which is in accordance with the more pronounced formation of randomly oriented BCO on IBAD”. Actually, the more pronounced formation of randomly oriented BCO on IBAD was observed only for one composition, i.e. x=0. I asked some clarifications about sentences at lines 125-127 and the related Figure 1. The authors replied me that the evidences to be clarified are due to some dirt in the lens of the microscope. Therefore, is such figure acceptable? Could the authors insert this explanation in the text? Otherwise, it is not clear the origin of the observed black spots. Moreover, I am not convinced that the substrate granularity can explain the streaks visible in the image related to Ni5W sample, in fact the grain visible in the figure has a different shape and smaller dimension with respect to the streaks.

Round 3

Reviewer 2 Report

Dear Authors,

thank you very much for the patience with my continual comments.

I consider that the manuscript is now acceptable for publication.